# SPLATTING-BASED MOTION CONTEXT ENCODING FOR DEEP VIDEO COMPRESSION

## ABSTRACT

Recent video compression studies aim to compress videos in a more optimal space using deep neural networks. Most of them employ a strategy where they use motion information to warp the previous frame to align with the current frame, and then only compress the information newly appearing in the current frame. While this enhances the compression efficiency of each frame, additional bits are required to compress the motion information alongside it. In this paper, we explore a methodology that improves motion compression by warping previous motions just like frames. However, within the traditional backward warping-based framework, a dilemma arises where the decoded motion is needed to warp the reference motion. To solve this problem, we propose a forward warping-based framework for video compression called SVC (Splatting-based Video Compression). While SVC offers the advantage of enabling the use of motion context, forward warping has several issues compared to backward warping and we propose additional tricks to address these challenges. Intensive experiments on the UVG, HEVC, and MCL-JCV benchmarks demonstrate that motion context encoding through SVC is indeed more effective compared to various methods based on backward warping, including traditional codecs.

## 1 INTRODUCTION

Video compression is a well-established area of research that seeks to represent video content with fewer bits without compromising its visual quality. The majority of existing video compression techniques, such as H.264 (Wiegand et al., 2003), H.265 (Sullivan et al., 2012), and H.266 (Bross et al., 2021), aim to increase the compression rate by compressing only the remaining information that is not present in the previous frame. In this context, the frame to be compressed is called the *current frame*, and the frame referred to for removing redundancy is called the *reference frame*. To achieve this, most of the existing techniques follow a three-step process. The first step is **motion estimation**, which involves matching the pixels that exist simultaneously in the current and previous frames. The second step is **motion compensation**, which involves rearranging the pixels of the reference frame using various warping methods so that the reference frame can be aligned with the current frame. Subsequently, only the newly added information in the current frame is encoded through **redundancy reduction** step, instead of compressing each frame independently.

The recent application of deep learning for image and video compression has shifted the compression domain from the conventional Discrete Cosine Transform (DCT) space to a more optimal domain learned end-to-end from large datasets. Furthermore, deep learning-based estimators have replaced traditional block-based motion vectors with dense optical flows (Dosovitskiy et al., 2015; Ranjan & Black, 2017; Sun et al., 2018). These advancements have greatly improved each step of the video compression process and demonstrated performance surpassing that of hand-crafted codecs. Further details on this topic are available in Section 2.

This paper diverges from prior works that mainly have focused on reducing redundancy between adjacent frames and instead concentrates on *reducing the redundancy between adjacent optical flows* caused by inertia and we call this **motion context encoding**. To minimize redundancy in images, the reference frame needs to be warped to align with the current frame, for which the optical flow map from the current frame to the reference frame is required. This causes the necessity of additional bits for the optical flow map. Fortunately, in the case of motion context encoding, the required

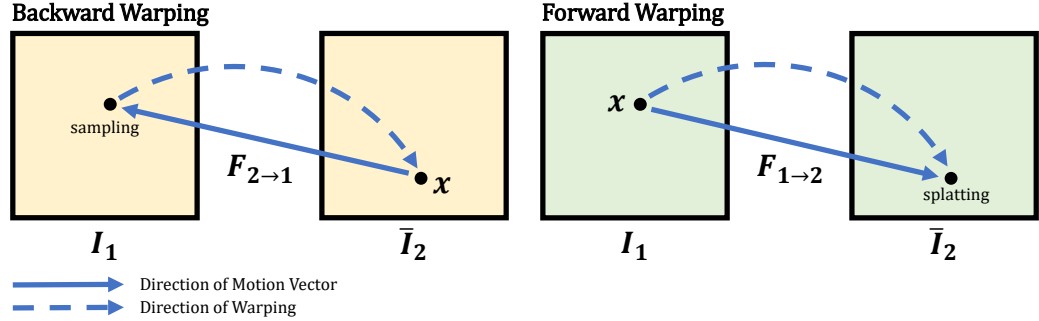

Figure 1: Backward Warping and Forward Warping.

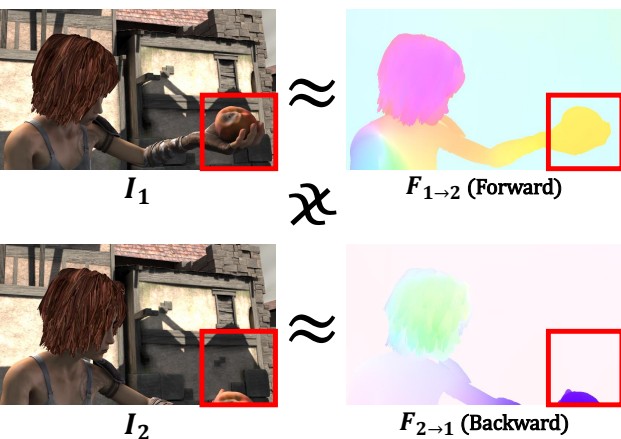

Figure 2: Edge alignment of the source image and optical flow.

motion vector for warping the reference optical flow map is the optical flow map itself, obviating the need for additional bits. However, the conventional backward warping-based paradigm presents a dilemma: to decode the current motion, the motion vector itself obtained from the decoding is needed to warp the reference optical flow map. In other words, the result of decoding is necessary for decoding. Detailed information on this issue is further addressed in Section 3.3.

To deal with this issue, we propose the use of forward warping instead of backward warping, as illustrated in Figure 1. In backward warping, since each motion vector corresponds to a pixel in the current frame, the target pixel is filled by sampling the pixel value at the location indicated by the motion vector. In contrast, in forward warping, each motion vector corresponds to a pixel in the reference frame, and these pixels are directly moved to the target position indicated by the motion vector to fill the target frame. By using forward warping, it becomes possible to warp the previous optical flow without problems, reducing redundancy between adjacent motions. This will be discussed in more detail in Section 3.3. Furthermore, unlike backward motion vectors, forward motion vectors are aligned with the reference frame (i.e. share the edges as shown in Figure 2), which allows for more efficient compression by utilizing redundancy between the optical flow map and the reference frame. Recently, an effective forward warping method called **splatting** has been introduced in the field of video frame interpolation (Niklaus & Liu, 2020; Niklaus et al., 2023).

In this context, we propose "splatting-based video compression (SVC)" that utilizes forward warping (splatting) to increase the compression efficiency of motion vectors. Unfortunately, unlike backward warping, splatting has some limitations such as the possibility of multiple pixels overlapping at the same location or creating holes. Therefore, this paper focuses on two main points. One is to minimize the aforementioned issues that forward warping has compared to backward warping. The other is to maximize the benefits of motion context encoding which is made possible through for-

ward warping. To achieve these, we first propose several methods to ensure that the forward warping module can achieve performance comparable to backward warping. Next, we propose a method to maximize the effect of motion context encoding.

## 2 RELATED WORK

While several traditional codecs such as H.263, H.264 (Wiegand et al., 2003), H.265 (Sullivan et al., 2012), H.266 (Bross et al., 2021), AV1 (Chen et al., 2018) have been well-tuned and continue to be widely used, there is still a need for efficient and fast video compression methods that can match the quality of these legacy standards. Recent advances in machine learning (ML) have shown promise in achieving more efficient and flexible video compression.

One possible approach is to skip frames and then interpolate them back (Wu et al., 2018; Djelouah et al., 2019; Park & Kim, 2019). Unlike the indirect methods, Ballé et al. (2016) and Ballé et al. (2018) proposed end-to-end frameworks for image compression, leading to the emergence of various studies aiming to adapt these for video compression. Some works tried to (Lu et al., 2019; Liu et al., 2019; 2020) replace each part of the traditional codec with deep learning-based modules, allowing for the exploration of a more optimal compression domain. Thanks to these studies, the deep video compression area has also established a standardized framework that follows the three stages of motion estimation, motion compensation, and redundancy reduction, just like traditional codecs.

Subsequently, various papers have extended different aspects based on this baseline. Most notably, Agustsson et al. (2020) proposed a method of modeling motion using scale space flow, and various attempts based on this method have emerged (Yang et al., 2020b; Rippel et al., 2021). An approach using RNN or ConvLSTM without using CNN has also emerged (Yang et al., 2020a; Golinski et al., 2020). Hu et al. (2021) proposed Feature-space Video Coding (FVC) that uses warping and deformable convolutions in a smaller feature space and Habibian et al. (2019) introduced an approach using a 3D autoregressive entropy model. On the other hand, Li et al. (2021) suggested a framework called Deep Contextual Video Compression (DCVC) that concatenates features instead of simple subtraction of the reference image to find more optimal redundancy reduction and this work is our baseline.

Various mode prediction methods have been proposed that incorporate many of the tricks used in traditional codecs (Hu et al., 2022), and generative models are also being studied (Mentzer et al., 2022a; Yang et al., 2021; Ho et al., 2022). Recently, methods using transformers, which move away from convolutional operations, have been proposed (Mentzer et al., 2022b).

Research has also been actively conducted to efficiently compress not only images but also motion information. Lin et al. (2020) employed a method that predicts the next motion using decoded previous motions across multiple frames. Taking it a step further, Rippel et al. (2021) proposed a strategy to store only the residuals of the optical flow. However, since they utilize the unwrapped reference flow rather than aligning the reference flow to the current flow through warping, redundancy may not be effectively removed. Thus, in this paper, just as warping the reference frame in images has been proven effective, we propose a method that warps the reference flow to align it with the current flow.

## 3 PROPOSED METHOD

### 3.1 PRELIMINARY

Let us consider compressing the $t$-th frame $I_t$ from a video that consists of $N$ frames $\{I_1, I_2, \cdots, I_N\}$. Most video compression methods aim to store only the newly appearing information in the current frame $I_t$ by excluding the information that is already present in both the current frame and the previously decoded reference frame $\hat{I}_{t-1}$. The reason for using the decoded frame $\hat{I}_{t-1}$ instead of the original frame $I_{t-1}$ is that the decoder cannot access the original frame.

To achieve this, most of the previous works obtain the warped frame $\overline{I}_t$ from the decoded reference frame $\hat{I}_{t-1}$ using backward warping $\overleftarrow{\omega}$ based on the decoded backward motion vector (optical flow)

$\hat{F}_{t \to t-1}$ from $I_t$ to $\hat{I}_{t-1}$, as follows (Please refer to Equation 5 for the mathematical definition of $\overleftarrow{w}$.).

$$\overline{I}_t = \overleftarrow{w}(\hat{I}_{t-1}, \hat{F}_{t \to t-1}) \tag{1}$$

Various methods can be used to remove redundancy between $\overline{I}_t$ and $I_t$, but typically, the residual information $\Delta I_t$ is obtained by subtracting $\overline{I}_t$ from $I_t$ as Equation 2.

$$\Delta I_t = I_t - \overline{I}_t \tag{2}$$

Then $\Delta I_t$ and the backward optical flow map $F_{t \to t-1}$ are encoded and transmitted. At the decoder end, the decoded reference frame $\hat{I}_{t-1}$, the decoded optical flow map $\hat{F}_{t \to t-1}$, and the decoded residual frame $\Delta \hat{I}_t$ are provided. The restored frame $\hat{I}_t$ is then obtained through the following process.

$$\overline{I}_t = \overleftarrow{w}(\hat{I}_{t-1}, \hat{F}_{t \to t-1}) \tag{3}$$

$$\hat{I}_t = \Delta \hat{I}_t + \overline{I}_t \tag{4}$$

## 3.2 BACKWARD WARPING AND FORWARD WARPING

Let us consider two frames, $I_0$ and $I_1$, and warp $I_0$ to align with $I_1$. For backward warping, we need backward motion $F_{1 \to 0}$ from $I_1$ to $I_0$, and we rearrange the pixels as follows.

$$\overline{I}_1(\boldsymbol{x}) = \overleftarrow{w}(I_0, F_{1 \to 0})(\boldsymbol{x}) = I_0(\boldsymbol{x} + F_{1 \to 0}(\boldsymbol{x})) \tag{5}$$

In Equation 5, if $\boldsymbol{x} + F_{1 \to 0}(\boldsymbol{x})$ does not point to a grid location of $I_0$, the pixel value is sampled using bilinear interpolation from the closest four pixels.

In the case of forward warping, we utilize the average splatting proposed in (Niklaus & Liu, 2020). To do this, we first define summation splatting as follows.

$$\beta(\boldsymbol{u} = [u_i, u_j]) = \max(0, 1 - |u_i|) \cdot \max(0, 1 - |u_j|) \tag{6}$$

$$\overrightarrow{w}_{sum}(I_0, F_{0 \to 1})(\boldsymbol{x}) = \sum_{\forall \boldsymbol{p} \in I_0} \beta(\boldsymbol{x} - (\boldsymbol{p} + F_{0 \to 1}(\boldsymbol{p}))) \cdot I_0(\boldsymbol{p}) \tag{7}$$

Then the average splatting $\overrightarrow{w}_{avg}$ is defined as follows (the notation for x is omitted because it is an independent operation for all locations x in Equation 8. $\mathbf{1}$ is an array consisting of ones and has the same resolution as $I_0$.).

$$\overline{I}_1 = \overrightarrow{w}_{avg}(I_0, F_{0 \to 1}) = \frac{\overrightarrow{w}_{sum}(I_0, F_{0 \to 1})}{\overrightarrow{w}_{sum}(\mathbf{1}, F_{0 \to 1})} \tag{8}$$

## 3.3 MOTION CONTEXT ENCODING

The main purpose of this paper is to minimize redundancy not only between adjacent images but also between adjacent optical flow maps. To compress the current motion vector $F_{t \to t-1}$, a reference motion vector $F_{t-1 \to t-2}$ is required. Therefore, two reference frames $\hat{I}_{t-1}$ and $\hat{I}_{t-2}$ are needed to compress $I_t$ using this method. We can obtain the two motion vectors using a motion estimation network $ME$ as follows.

$$F_{t \to t-1} = ME(I_t, \hat{I}_{t-1}), \quad F_{t-1 \to t-2} = ME(\hat{I}_{t-1}, \hat{I}_{t-2}) \tag{9}$$

Then we can get the warped reference motion vector $\overline{F}_{t\to t-1}$ by backward warping $F_{t-1\to t-2}$ using $F_{t\to t-1}$ as follows (We should warp according to the decoded flow $\hat{F}_{t\to t-1}$, but since it's before decoding, let's assume we use $F_{t\to t-1}$ for now.).

$$\overline{F}_{t\to t-1} = \overleftarrow{\omega}(F_{t-1\to t-2}, F_{t\to t-1}) \tag{10}$$

$$\Delta F_{t\to t-1} = F_{t\to t-1} - \overline{F}_{t\to t-1} \tag{11}$$

This part can be confusing, but simply put, $F_{AB}$ has no direct relationship with $I_B$ and is just the same image with a different modality that shares the same structure with $I_A$, such as edges (as shown in Figure 2).

Obtaining $\overline{F}_{t\to t-1}$ through backward warping is not a problem in the encoding process, but it causes a dilemma in the decoding process. During decoding, we cannot use the original flow $F_{t\to t-1}$, so we have to use the decoded flow $\hat{F}_{t\to t-1}$; however, this is also unobtainable before decoding (see Equations 12 and 13.).

$$\overline{F}'_{t\to t-1} = \overleftarrow{\omega}(F_{t-1\to t-2}, \hat{F}_{t\to t-1}) \tag{12}$$

$$\hat{F}_{t\to t-1} = \Delta F_{t\to t-1} + \overline{F}'_{t\to t-1} \tag{13}$$

A simple solution to this dilemma is to use forward motion instead of backward motion as follows.

$$F_{t-1\to t} = ME(\hat{I}_{t-1}, I_t), \quad F_{t-2\to t-1} = ME(\hat{I}_{t-2}, \hat{I}_{t-1}) \tag{14}$$

Then, we can obtain $\overline{F}_{t-1\to t}$ by forward warping $F_{t-2\to t-1}$ using $F_{t-2\to t-1}$ itself as follows.

$$\overline{F}_{t-1\to t} = \overrightarrow{\omega}_{avg}(F_{t-2\to t-1}, F_{t-2\to t-1}) \tag{15}$$

$$\Delta F_{t-1\to t} = F_{t-1\to t} - \overline{F}_{t-1\to t} \tag{16}$$

As a result, at the decoding stage, we only need the reference motion $F_{t-2\to t-1}$ which can be easily obtained from the two reference frames instead of the current motion $F_{t-1\to t}$. Therefore we can decode $\hat{F}_{t-1\to t}$ without any problems as shown in Equations 17 and 18.

$$\overline{F}_{t-1\to t} = \overrightarrow{\omega}_{avg}(F_{t-2\to t-1}, F_{t-2\to t-1}) \tag{17}$$

$$\hat{F}_{t-1\to t} = \Delta F_{t-1\to t} + \overline{F}_{t-1\to t} \tag{18}$$

## 3.4 BASELINE ARCHITECTURE

The baseline model used in this paper is fundamentally based on DCVC (Li et al., 2021). In other words, instead of simply subtracting the current and reference frames as in Equation 2, we adopt an approach that encodes the context obtained from the features extracted from both frames. Consequently, the warping operations (Equations 5 and 8) involve warping their feature maps rather than directly warping the image or flow map. However, for clarity, we will omit this detail in the notation. In addition, the obtained context is encoded via Hyperprior (Ballé et al., 2018) method. Moreover, we improve the baseline by applying several effective tricks recently proposed. The following changes are applied to our baseline.

**Checkerboard Context Module.** To utilize the mutual information between feature vectors within a single frame, DCVC adopts the autoregressive encoding (Minnen et al., 2018) approach. However, this method has the drawback of slow computation speed since it cannot process all pixels in a frame simultaneously in parallel. Therefore, instead of this module, we use the checkerboard context model (He et al., 2021), which can refer to the already decoded surrounding pixels while enabling parallel processing.

**Multiscale Feature.** DCVC only uses features extracted from a single scale to extract context. However, it has been proven that utilizing multi-scale features is more effective in various research

fields, including video compression (Li et al., 2022). Therefore, our baseline also uses features extracted from five different scales, each of which is warped to obtain context.

**Rate Controllable EASN.** DCVC uses Generalized Divisible Normalization (GDN)(Ballé et al., 2016) as the activation function for the hyperprior encoder and decoder. Instead of GDN, we use EASN (Shin et al., 2022) which has been proposed and shown to be more stable and perform better than GDN as the activation function in our model. Additionally, we improve the model by incorporating a learnable vector table to EASN and passing the sampled vector along with the encoded features. This allows us to adjust the rate parameter to control the rate-distortion tradeoff without retraining the model.

### 3.5 MINIMIZING THE DRAWBACKS OF FORWARD WARPING

As mentioned in Section 1, although forward warping has the advantage of enabling motion context encoding, it has some problems compared to backward warping. One of the most typical issues is that two or more pixels can be projected onto the same location, or no pixels may be projected on somewhere else, resulting in holes. Fortunately, we do not directly calculate the residual through subtraction like Equation 2, but instead extract context like DCVC (Li et al., 2021). This allows us to have some robustness against holes and pixel overlapping. However, forward warping still has some drawbacks compared to backward warping, and there is room for improvement. In this section, we propose three effective tricks to minimize the performance degradation of the proposed forward warping module compared to traditional backward warping.

**Reference Image Guidance.** As shown in Figure 2, unlike backward motion $F_{t \to t-1}$, forward motion $F_{t-1 \to t}$ shares the structure with the reference frame $I_{t-1}$, which leads to mutual information. Therefore, we provide $I_{t-1}$ as guidance to the motion encoder and decoder, enabling them to omit the flow edge information.

**Flow Reversing.** The problems of overlapping pixels and holes can also interfere with the smooth flow of gradients during end-to-end learning. For example, if all pixels are projected to one point and all other locations are holes, the gradient flows only through one pixel. Therefore, instead of directly splatting the reference image (or feature map in the case of the DCVC baseline) like Equation 8, we obtain a pseudo backward flow $\tilde{F}_{t \to t-1}$ by reversing and splatting the reversed optical flow $-F_{t-1 \to t}$ like (Niklaus et al., 2023), as follows.

$$\tilde{F}_{t \to t-1} = \overrightarrow{\omega}_{avg}(-F_{t-1 \to t}, F_{t-1 \to t}) \tag{19}$$

Then we can get the forward warped result by backward warping with $\tilde{F}_{t \to t-1}$ as follows.

$$\overline{I}_t = \overrightarrow{\omega}_{rev}(\hat{I}_{t-1}, F_{t-1 \to t}) = \overleftarrow{\omega}(\hat{I}_{t-1}, \tilde{F}_{t \to t-1}) \tag{20}$$

Equation 20 is essentially forward warping, but the operation actually applied to $\hat{I}_{t-1}$ is backward warping, which allows for smooth gradient flow.

**Gradient Stopping.** While flow reversing enables smooth gradient flow for the images or feature maps being warped, the gradient of Equation 19 can still affect the optical flow $F_{t-1 \to t}$ and performance of motion estimation. To prevent it, we apply a gradient stop $S[\cdot]$ to the second term in Equation 19, and only allow the gradient to flow through the first term as follows.

$$\tilde{F}_{t \to t-1} = \overrightarrow{\omega}_{avg}(-F_{t-1 \to t}, S[F_{t-1 \to t}]) \tag{21}$$

### 3.6 MAXIMIZING THE ADVANTAGES OF MOTION CONTEXT ENCODING

In Section 3.3, we obtain the reference $\overline{F}_{t-1 \to t}$ of current motion $F_{t-1 \to t}$ by warping $F_{t-2 \to t-1}$ using Equation 17. However, as mentioned in Section 3.5, forward warping operation has several limitations. Fortunately, unlike in the case of warping images, there exists an excellent alternative that has a strong correlation with $F_{t-1 \to t}$ and does not require warping. It is the reverse of $F_{t-1 \to t-2}$.

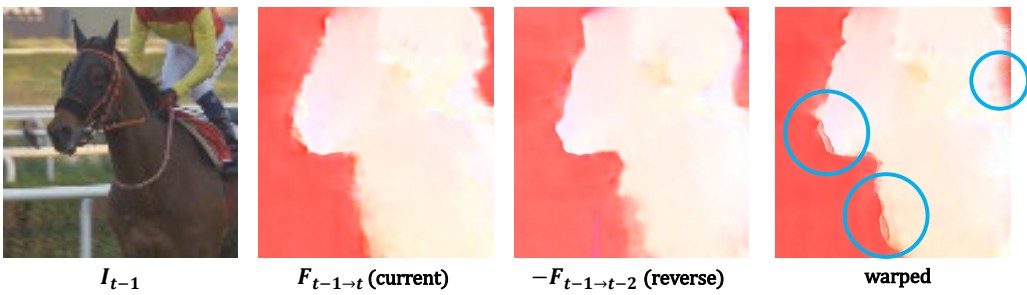

$$I_{t-1} \qquad F_{t-1 \to t} \text{ (current)} \qquad -F_{t-1 \to t-2} \text{ (reverse)} \qquad \text{warped}$$

Figure 3: Reference candidates of current motion.

$$\overline{F}_{t-1 \to t} = -F_{t-1 \to t-2} \qquad (22)$$

As can be seen in Figure 3, the reference obtained through warping (4th image of Figure 3) suffers from artifacts due to pixel overlapping compared to $-F_{t-1 \to t-2}$ (3rd image of Figure 3).

### 3.7 LOSS FUNCTION

We use Rate-Distortion (RD) trade-off loss following (Ballé et al., 2016; 2018) to train the proposed model.

$$L = R + \lambda D = R + \lambda d(I, I^*), \qquad (23)$$

where $R$ denotes the bit rate and $d$ denotes distortion function. We use mean squared error (MSE) loss as d.

## 4 EXPERIMENTS

### 4.1 SETTINGS

To demonstrate the effectiveness of the contribution and main idea of this paper, we test various versions of our method and some comparison algorithms on UVG (Mercat et al., 2020), HEVC (Sullivan et al., 2012), and MCL-JCV (Wang et al., 2016) datasets. The UVG and MCL-JCV datasets consist of frames of size $1920 \times 1024$, while the HEVC dataset is Composed of four classes B, C, D, E with various sizes. All test videos are center-cropped to have a width and height that are multiples of 64, and the test GOP (Group Of Pictures) size is fixed at 12.

### 4.2 TRAINING

**Training Dataset.** We use the Vimoe-90K (Xue et al., 2019) septuplet dataset consisting of 89,800 video clips with 7 frames of size $448 \times 256$ and. For training, we randomly crop each frame to size $256 \times 256$ and use random horizontal/vertical flips and rotations for data augmentation.

**Implementation Detail.** We implement the proposed model using the PyTorch (Paszke et al., 2019) library and utilize the Cupy (Okuta et al., 2017) library for splatting operation. We use the AdamW (Loshchilov & Hutter, 2017) optimizer with a batch size of 4 and a fixed learning rate of $10^{-4}$. We train the model for a total of 46 epochs, where the first 6 epochs are dedicated to training the motion encoder, the next 6 epochs are dedicated to training the image encoder, and the remaining 34 epochs are dedicated to training the entire model simultaneously.

### 4.3 ABLATION STUDY

This section verifies whether the contributions of the paper, including forward warping framework and the tricks introduced in Section 3.5 and Section 3.6, are actually effective.

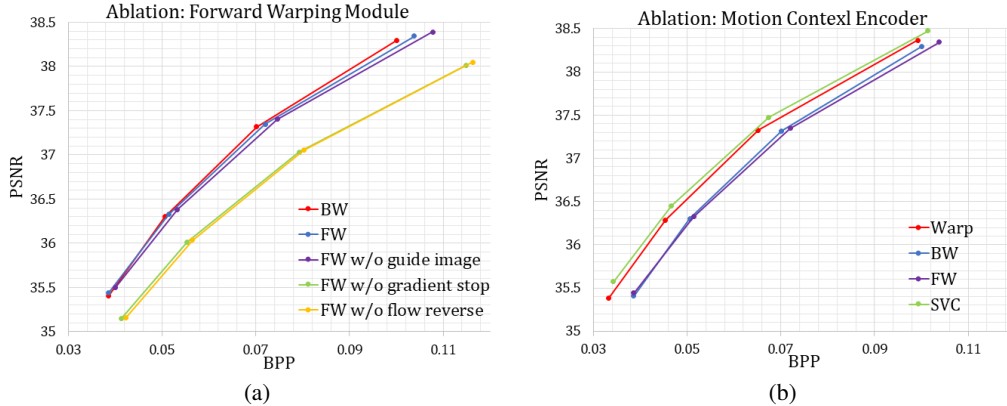

Figure 4: (a) Ablation study on forward warping module (BW means backward warping and FW means forward warping.). (b) Ablation study on motion context encoder.

Figure 4 (a) presents the results of the experiments verifying the tricks introduced in Section 3.5. All versions in Figure 4 (a) are the results without applying motion context encoding. As mentioned, the performance of the forward warping operation itself is inferior to that of the backward warping. To minimize this gap, the version that applies all three tricks in Section 3.5 is FW. It shows that our forward warping module has succeeded in approaching the performance of backward warping. In particular, gradient stop and flow reverse can be concluded as essential tricks for stable training, therefore they are necessary for forward warping.

Figure 4 (b) demonstrates that the version that reverses the backward flow (SVC) performs better than the version that warps the reference motion vector (Warp) as introduced in Section 3.6. In addition, despite the drawback of forward warping, the attempt to reduce the redundancy of motion vectors through motion context encoding shows better performance than the backward warping-based baseline, proving the effectiveness of the main idea of this paper.

## 4.4 COMPARISON

We compare our SVC with x265 (Sullivan et al., 2012) veryslow preset, VTM (Bross et al., 2021) and HM codec with low delay mode, and DCVC (Li et al., 2021), a representative deep learning-based video compression method. Most of the previous deep learning-based video compression papers have often compared their models with other state-of-the-art models without standardizing the intra mode, which cannot be considered a fair comparison. Our model has two characteristics related to intra frame coding. The first is that it can start compression from the third frame because it requires two reference frames. The second is that we do not propose a fixed intra method, so any model can be plugged in without any issues. Therefore if we are comparing its performance with another codec $\mathcal{C}$, the first and second frames are compressed with $\mathcal{C}$. This method allows for a fair comparison with any other codec. Figure 5 shows that the proposed model, SVC, clearly outperforms x265 veryslow, DCVC, and HM, and performs similarly or even slightly better for large $\lambda$ values compared to VTM.

## 5 LIMITATION

The method proposed in this paper is one of many possible forward warping methods. It is not optimal and there are still many issues such as unstable training to be addressed compared to backward warping. We hope that the forward warping-based paradigm will continue to evolve through future research in this field.

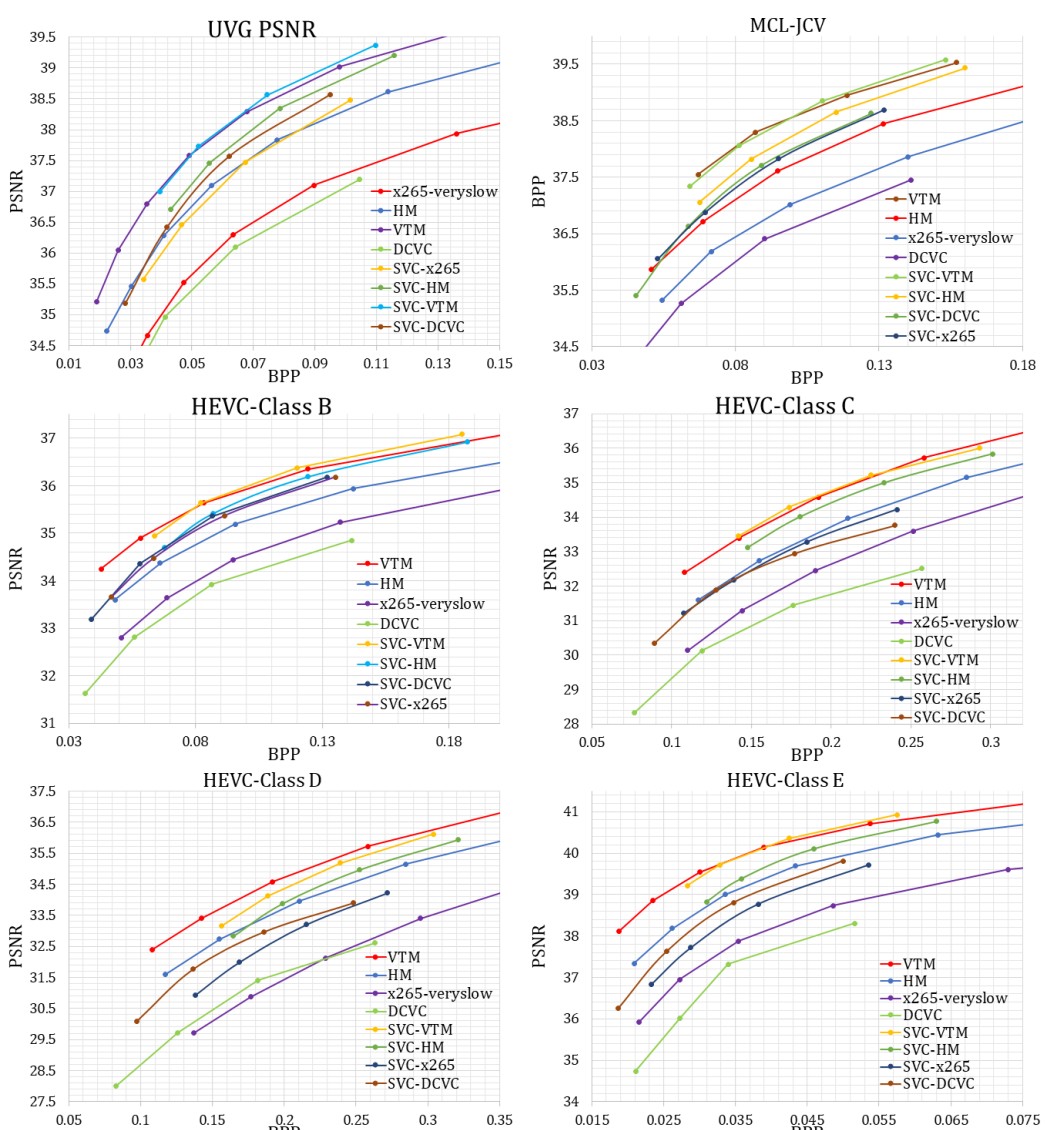

Figure 5: The comparisons of rate distortion curves on various test datasets.

## 6 CONCLUSION

In this paper, we explore the dilemma arising from directly applying the traditional backward warping, used for compressing frames, to motion. To circumvent this issue, we propose a forward warping-based method. This allows the reference optical flow map to align with the current motion, enabling more efficient motion compression. However, some inherent challenges associated with forward warping become evident. We therefore introduce several tricks to address these challenges. Through experimentation, we confirm that motion context warping via forward warping is more effective than existing methods. We anticipate that simply reversing the warping direction, as demonstrated, could not only benefit video compression but also serve as a game changer in other motion-related video processing fields.

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
