# OpenReview forum: "Splatting-based Motion Context Encoding for Deep Video Compression"
_ICLR.cc/2024/Conference — ICLR 2024 Conference Withdrawn Submission_

### Official Review · Reviewer_5LRL · 2023-10-27

**Soundness:** 2 fair
**Presentation:** 1 poor
**Contribution:** 2 fair
**Rating:** 3
**Confidence:** 5

**Summary:**

This paper introduces SVC (Splatting-based Video Compression), a forward warping-based framework for video compression. SVC improves motion compression by warping previous motions similarly to frames, enabling the use of motion context.

**Strengths:**

The paper introduces a phenomenon to analyze backward warping and forward warping, which may bring new insight into solving the motion prediction for video compression and sound technical.

**Weaknesses:**

However, the techniques used are already proposed by other methods, such as splatting the reversed flow from Niklaus et al., Gradient stopping from learned image compression, so the innovation of this paper is conceptual without solid supports. Detailed weaknesses are shown below:

1) As shown in Fig. 2, if $I_1$ changes into $I_2$, and $I_2$ changes into $I_1$, which means Backward warping may be better than forward warping?

2) In video compression decoding, use the reference frame and motion to predict the current frame, so forward warping needs to use the current frame and motion to predict the reference frame.

3) Authors may explain the difference between techniques used in this paper and the original ones, such as splatting the reversed flow from Niklaus et al.

4) The insufficient investigation of Learned Video Compression, such as experimental comparison with them, and the comparison with motion-free video compression (latent space).

5) It is hard to distinguish some lines in Fig. 5 due to the similar colors. The superiority of forward warping is not always better than backward in Fig 4 and Fig 5. It is not clear that the performance of the method when BPP is high, such as 0.5 - 1.0.

6) Overall, it is hard to read and understand this paper.

**Questions:**

See Weaknesses.

---

### Official Review · Reviewer_iK9f · 2023-10-31

**Soundness:** 2 fair
**Presentation:** 3 good
**Contribution:** 3 good
**Rating:** 3
**Confidence:** 4

**Summary:**

This paper focuses on improving the motion estimation module in video compression, by using forward warping instead of the conventional backward warping approach. The goal is to reduce the redundancy between adjacent optical flows to save the bit cost in video compression. While in the scenario of video coding there would be a dilemma if we want to freely produce the target optical flow using the previous one, the forward warping approach could bypass this problem and "splat" pixels in the previous optical flow to approximate the target optical flow. The experiments in this paper demonstrate the proposed could be plugged to both traditional video codecs like HM and VTM, and the neural video codecs like DCVC, bringing gains especially in neural video codec.

**Strengths:**

The motivation of this paper is clear: use the redundancies between the previous optical flow and the target one so that the coding cost of motion information would be reduced. And the solution in this paper makes sense in general, i.e., using forward warping and some postprocessing operations to produce effective motion context. The experimental results (Figure 5) seems to be encouraging: the proposed approach can be plugged to both traditional codecs and neural video codecs.

**Weaknesses:**

Although the idea of forwarding warping with splatting-based motion context encoding makes sense in concept, I have several concerns regarding the current version.

(1) Many important experimental settings should be elaborated. In the field of video compression, it is very important to figure out the coding configurations. Some papers tried to keep consistent by always using a single reference frame for prediction (low-delay P mode, with the shortest latency), and some papers may use multiple reference frames for unidirectional or bidirectional prediction (which would usually take much longer time for decoding/encoding). This paper focuses on unidirectional inter-prediction and states in the paper like "compare with VTM and HM with low delay mode". However, the proposed method in this paper involves using the previous optical flow to generate motion context, which inevitably takes the information from at least two reference frames. But those baselines compared in this paper are ambiguous in reference configuration. On the one hand, DCVC is a neural video codec using a single reference frame. It would be unfair to directly compare with it. On the other hand, it is important to include the configurations of VTM and HM in order to convince the readers that the comparisons are fair enough.

(2) While the enhanced DCVC in this paper performs much better than the original DCVC, I am unsure whether the gains come from the mechanism of forward warping itself. There are several enhancement tricks also used in this paper, like multiscale features and checkerboard context module. Some of them have been demonstrated to be able to improve the performance significantly [Li et al.,Hybrid spatial-temporal entropy modelling for neural video compression].

(3) Considering an extra motion context encoder is deployed in this paper to refine the artifacts of forward warping, which is shown to be very effective, there should be some detailed introductions of this motion context encoder, including the number of network parameters and the computational complexity.   In addition, perhaps some technical improvements can be further applied such as using softmax splatting instead of the average splatting.

(4) Splatting-based motion synthesis was first proposed in previous papers of video frame interpolation, Niklaus et al., 2021 & 2023. I understand there would be a unique problem if we want to make the similar idea work in the task of video compression, but a previous paper has already involves a similar idea in video compression, which should be an important reference or at least deserve some discussions in this work [Ref1].
[Ref1] Versatile Learned Video Compression. Feng et al.. Arxiv, 2021.

**Questions:**

See the abovementioned weaknesses. Besides, could the author introduce in detail about how the proposed method was plugged in traditional codecs like VTM and HM?

---

### Official Review · Reviewer_eU62 · 2023-11-01

**Soundness:** 3 good
**Presentation:** 2 fair
**Contribution:** 2 fair
**Rating:** 3
**Confidence:** 5

**Summary:**

This paper presents a forward warping-based framework for video compression to improves motion compression. Forward warping has several issues compared to backward warping and the authors propose additional tricks to address these challenges.

**Strengths:**

This paper presents a forward warping-based framework for video compression to improves motion compression. Forward warping has several issues compared to backward warping and the authors propose additional tricks to address these challenges.

**Weaknesses:**

However, this paper has obvious flaws and the experimental results are insufficient for publication requirements in my perspective. Specifically, I have the following concerns:

1. The main idea of motion context coding proposed in the paper is to use motion information between frames $I_{t-1}$ and $I_{t-2}$ to predict the motion of the current frame. This technique has been explored in previous works [1] [2].

[1] M-LVC: Multiple Frames Prediction for Learned Video Compression

[2] Deep Incremental Optical Flow Coding for Learned Video Compression

Therefore, there is not enough novelty and I do not see significant differences between the proposed method and previous works. Additionally, the authors did not compare their approach with these methods.

2. In terms of experiments, it would be beneficial if the authors compared their work with recent approaches such as AlphaVC, DCVC-HEM, DCVC-DC, etc. Furthermore, comparing with DCVC may not be fair enough since the proposed method utilizes multiscale features and other techniques that were not included in the original DCVC paper.

**Questions:**

In Figure 5 of the paper, the colors of the lines are confusing, making it difficult to distinguish. It is recommended to make modifications.

---

### Official Review · Reviewer_Huhc · 2023-11-05

**Soundness:** 2 fair
**Presentation:** 2 fair
**Contribution:** 2 fair
**Rating:** 3
**Confidence:** 4

**Summary:**

This paper introduces a neural video compression framework that leverages motion context encoding by warping the previous motions. For this purpose, the paper utilizes forward warping and addresses two major challenges associated with forward warping that are not present in backward warping. Through experiments on various benchmarks, the paper demonstrates that integrating the proposed motion context encoding with established neural video compression methods significantly enhances rate distortion (RD) performance. This suggests that the proposed module effectively minimizes redundancies not only between frames but also between adjacent optical flow maps.

**Strengths:**

- The paper is easy to read.
- The use of forward warping is well-motivated compared to backward warping.
- Integration of the proposed method with existing neural video compression models shows consistent RD performance improvements.

**Weaknesses:**

- The scope of the paper is narrow, with a primary focus on forward warping for compressing optical flow maps. Thus, the authors mainly compare the effectiveness of their proposed forward-warping method with the integration of established neural video compression methods, as shown in Figure 5. However, as outlined in the Related Work section, recent advances in neural video compression typically do not employ optical flow maps in pixel space. Therefore, the argument for reducing redundancy between adjacent optical flows, as opposed to recent advances in neural video compression methods, is not fully convincing without demonstrable comparable effectiveness.

- Editorial comment: In Figure 5, to avoid confusion, it is recommended to use a consistent color scheme with the other figures. Utilizing similar colors for both the integrated and non-integrated versions of the proposed method could more clearly indicate the effectiveness of the proposed method.

- The paper lacks measurements of computational time. The computational overhead introduced by the additional warping may affect both the training time and the decoding performance. It is crucial to provide comprehensive studies on the encoding and decoding times in comparison with existing neural video compression methods.

**Questions:**

As mentioned in the weakness section, the adoption of forward warping compared to backward warping seems reasonable, but the adoption of optical flow for video compression seems limited. It is recommended that the authors clarify the necessity of the adoption of optical flow for video compression.